# Tsunami propagation kernel and its applications

Takenori Shimozono[1]

[1]The University of Tokyo, Tokyo, Japan

**Correspondence:** Takenori Shimozono (shimozono@coastal.t.u-tokyo.ac.jp)

**Abstract.** Tsunamis rarely occur in a specific area, and their occurrence is highly uncertain. Suddenly generated from their sources in deep water, they occasionally undergo tremendous amplification in shallow water to devastate low-lying coastal areas. Despite the advancement of computational power and simulation algorithms, there is a need for novel and rigorous approaches to efficiently predict coastal amplification of tsunamis during different disaster management phases, such as tsunami risk assessment and real-time forecast. This study presents convolution kernels that can instantly predict onshore waveforms of water surface elevation and flow velocity from observed/simulated wavedata apart from the shore. Kernel convolution involves isolating an incident-wave component from the offshore wavedata and transforming it into the onshore waveform. Moreover, unlike previously derived ones, the present kernels are based on shallow-water equations with a damping term and can account for tsunami attenuation on its path to the shore with a damping parameter. Kernel convolution can be implemented at a low computational cost compared to conventional numerical models that discretise the spatial domain. The prediction capability of the kernel method was demonstrated through application to real-world tsunami cases.

## 1 Introduction

Tsunamis pose a major threat to low-lying coastal areas worldwide. They are initiated by the rapid displacement of seawater, which is often triggered by earthquakes and/or submarine landslides. Given the initial water-surface displacement, we are currently able to simulate tsunami propagation from the source to coastal areas and then assess tsunami hazards on coastal communities with a practical level of accuracy. However, the occurrence of a tsunami is highly uncertain; thus, we need to prepare for potential tsunami hazards by considering many different source scenarios that are often based on scarce historical data. This involves performing numerous tsunami simulations that are often implemented by a two-step approach, i.e. deep-water tsunami modelling and coastal tsunami modelling. The first step predicts tsunami propagation from the source to coastal shelves; the linear shallow-water equations are solved on a relatively coarse grid. Then, the second step simulates coastal tsunami evolution using non-linear shallow-water equations discretised on a finer grid that can resolve coastal bathymetry. Although the two steps can be combined by grid refinement algorithms, the coastal tsunami simulation still requires high computational costs; this may limit the range of uncertainty covered by the scenarios. There is a demand for novel approaches to rapidly predict coastal tsunami evolution despite the growing computational power. Such methods will also contribute to the real-time prediction of coastal tsunami impacts from a tsunami waveform that is observed or predicted in deep water.

The coastal evolution of tsunamis has been analytically studied by many researchers using shallow-water equations because the vertical motion of seawater is small compared to the horizontal motion during the propagation phase. The main physics of wave transformation can be described by the linear shallow-water equations, while the non-linear effect is responsible for the wave distortion in shallow water (Carrier and Greenspan, 1958). When the wave height is large, relative to water depth, short-wavelength components exhibit wave breaking owing to non-linear wave distortion, which considerably dissipates the wave energy. However, most past tsunamis in world oceans are known to have propagated as non-breaking waves. Analytical solutions for non-breaking wave evolution over a uniform slope have been derived for incident transient waves in different forms, which allow us to predict tsunami run-up height on the shore by prototype incident waves (Synolakis, 1987; Tadepalli and Synolakis, 1994; Pelinovsky and Mazova, 1992). This approach can be further extended for non-uniform seabed profiles (Didenkulova et al., 2009) and narrow bays (Zahibo et al., 2006; Rybkin et al., 2014; Shimozono, 2016). The resulting run-up formulas provided valuable insights into the coastal evolution processes of tsunamis in different forms. However, actual tsunami waveforms are diverse and differ from these prototype waves. Moreover, the highest impact is not necessarily produced by a leading part of tsunamis, especially in cases of far-field sources. Therefore, the parametric formulas for specific wave types do not work directly for real-world tsunami problems.

A possible way to rapidly predict a real-world tsunami transformation is to use kernel representation of long wave propagation over a sloping coast. Coastal tsunami profile on a uniform slope can be expressed in the frequency domain as a product of a Bessel transfer function and an integral transform of the incident-wave profile (Synolakis, 1991). This, in turn, means that the tsunami waveform in the time domain results from a convolution of the incident-wave profile with a kernel function, which is the inverse transform of the Bessel transfer function. Kernel convolution enables us to predict a tsunami waveform given an incident-wave profile of any integral form. A bottleneck for this approach was that the Bessel transfer function could not be readily inverted into the time-domain kernel function. Madsen and Schäffer (2010) derived an approximate kernel for wave profile on the shore using asymptotic expressions of Bessel functions. More recently, the author derived a general kernel for wave profile at an arbitrary location on the slope, which yields an exact solution of shallow-water equations via the convolution with the incident-wave time series (Shimozono, 2020). There have been few attempts to apply kernel representation for the rapid prediction of tsunamis (Choi et al., 2011; Chan and Liu, 2012) because it still has some limitations for real-world applications. First, the real-world problem cannot be represented as the incident-wave boundary value problem. The observed or simulated tsunamis in deep water more or less contain reflected waves from the coast. In addition, real-world tsunamis are attenuated by different factors as they propagate to the shore. The kernel representation based on frictionless shallow-water equations does not account for the wave decay that becomes significant in shallow water.

As an extension of the author's earlier work (Shimozono, 2020), this paper presents a new kernel representation that has higher applicability to real-world tsunami problems. Unlike the previous ones derived under the incident-wave boundary condition, the presented kernel functions are formulated with the observed-wave boundary condition that may include a contribution of reflected waves from the coast. Moreover, the kernel is derived from shallow-water equations with a damping term. Therefore, a damping factor is incorporated into the kernel, which can be used to represent the tsunami decay on its path to the shore. The practical kernel works when an onshore tsunami profile needs to be predicted directly from observed or simulated

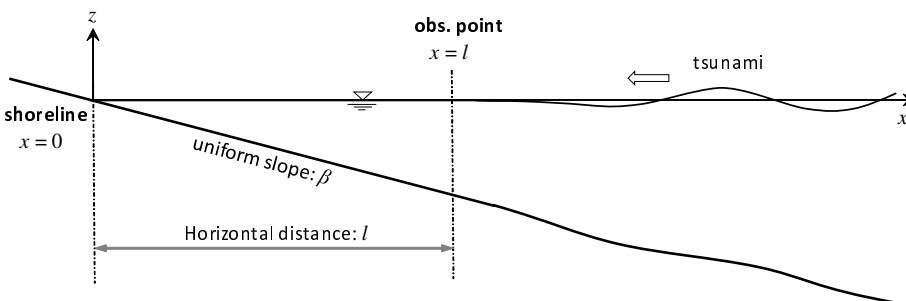

**Figure 1.** Schematic illustration of the boundary value problem.

wavedata apart from the shore. The next section presents the derivation of kernel functions for water surface elevation and flow velocity from the shallow-water equations under the observed-wave boundary condition. Subsequently, Section 3 presents an efficient numerical method of implementing kernel convolution. Section 4 demonstrates the predictive capability of the kernel method through its applications to 2011 Tohoku tsunami cases. Finally, Section 5 provides brief conclusions of this study.

## 2  Kernel functions

### 2.1  Kernel represenation

Let us consider one-dimensional propagation of a tsunami from deep ocean to coastal water, which is initially at the stationary state, as illustrated in Fig 1. We define the $x^*$-axis as the positive-seaward horizontal coordinate with an origin at the still water shoreline. There is a wave observation point at $x^* = l$, where water surface displacements driven by the incoming tsunami are continuously measured. The seabed slope landward of the observation point is assumed to be uniform, while the seabed on its seaside may have an arbitrary form as long as it smoothly connects to the landward uniform slope. The measured wave profile contains both landward-propagating waves from the source and seaward-propagating waves reflected from the shore. Our problem is to determine a coastal wave profile under the observed-wave boundary condition. This is different from the previous formalism in which coastal wave evolution is predicted under the incident-wave boundary condition by assuming a horizontal bed seaward of the boundary (Keller and Keller, 1964; Carrier, 1971; Synolakis, 1991).

Here, we use linear shallow-water equations to formulate the problem. Wave non-linearity becomes significant in shallow water, and the non-linear effect distorts the time and spatial axes. However, it is well known that the wave amplitude is not significantly affected by non-linearity unless the non-linear wave distortion leads to wave breaking (Carrier and Greenspan, 1958; Tuck and Hwang, 1972; Synolakis, 1991). The nonlinear shoreline motion can be readily derived from the linear solution via the hodograph transform, and the run-up height is unchanged from the linear case, if the boundary-value assignment is linearized (e.g. Pelinovsky and Mazova, 1992). Furthermore, the ocurrence of wave breaking, which limits the applicable range of the present approach, can be predicted as a breakdown point of the hodograph transform under the same condition. While the linearized boundary-value assignment potentially affects the run-up height and the wave breaking condition, nonlinear

modifications are minor as long as the ratio of wave amplitude to water depth is small at the boundary (Antuono and Brocchini, 2007, 2008). Therefore, the main process of practical interest can be described by the linear equations when we place the boundary in deep water. Another important fact, which has been often neglected in analytical models, is that a tsunami decays on its path to the shore by different factors. Therefore, we incorporate a damping term into the momentum equation to account for the tsunami decay. Accordingly, the damped propagation of tsunamis over a uniform slope can be described by the following equations:

$$\frac{\partial \eta^*}{\partial t^*} + \beta u^* + \beta x^* \frac{\partial u^*}{\partial x^*} = 0, \tag{1}$$

$$\frac{\partial u^*}{\partial t^*} + g \frac{\partial \eta^*}{\partial x^*} = d^*. \tag{2}$$

Here, the asterisk superscript represents a dimensional variable. Therefore, $t^*$ represents time, $\eta^*(x^*, t^*)$ is the water surface elevation above the still water level, $u^*(x^*, t^*)$ is the positive-seaward horizontal velocity, $\beta$ is the seabed slope, $d^*$ is the damping term, and $g$ is the gravitational acceleration.

The quadratic law is often used to model tsunami damping ($d^* \propto |u^*| u^*$), but this introduces non-linearity into the problem, which makes the analytical work infeasible. Instead, we employ the linear damping term as follows;

$$d^* = -\alpha^* u^*, \tag{3}$$

where $\alpha^* (\geq 0)$ is the damping parameter, which has a dimension of the reciprocal of time. The linear damping term introduces an exponential wave decay with the characteritic time corresponding to $\alpha^{*-1}$ into the non-dissipative solutuon (Davies et al., 2020). In virtue of its simple formulation, the linear model has been used to represent damped propagation of tsunamis in deep water. Many reserchers suggested the damping parameter of $\mathcal{O}(10^{-5})$ s$^{-1}$ for deep-water tsunami propagation through comparisons of the model and observations (e.g. Fine et al., 2013; Kulikov et al., 2014). However, the damping parameter could be much higher for neashore tsunami dissipation. Mazova et al. (1990) analytically derived a dissipative runup formula of monochromatic long waves over a uniform slope using the linear damping term. They suggested the $\alpha^*$ value of $\mathcal{O}(10^{-2})$ s$^{-1}$ in a typical situation of calculating tsunami run-up.

Tsunamis decay in shallow water by different factors, such as skin and form drag over seabed as well as local topography and coastal structures, the scales of which are much smaller than the tsunami wavelength. Here, we attempt to represent the tsunami decay due to different factors collectively by the linear damping term; thus, the parameter $\alpha^*$ works as an empirical parameter rather than a physical parameter. This treatment makes the resulting kernel representation to be a semi-empirical one. We will discuss the variability of this parameter through real-world tsunami applications in the later section.

The governing equations are non-dimensionalised by the following dimensionless variables;

$$x^* = xl, \quad t^* = t\sqrt{l/g\beta}, \quad \eta^* = \eta\beta l, \quad u^* = u\sqrt{g\beta l}, \quad \alpha^* = \alpha\sqrt{g\beta/l}. \tag{4}$$

Here, the spatial and time scales are chosen to be the horizontal slope length, $l$ and half of the wave travel time over the entire slope $\sqrt{l/g\beta}$, respectively. Namely, the horizontal dimensionless coordinate takes the value of $x = 0$ at the shoreline and $x = 1$ at the offshore boundary. The dimensionless time of $t = 2$ equals to the one-way travelling time from/to the boundary to/from the shore. Of note, the shallow water approximation requires $l \ll L_0/\beta$, where $L_0$ is the representative wavelength.

Substituting (4) into (1) and (2) yields the dimensionless forms of the governing equations:

$$\frac{\partial \eta}{\partial t} + u + x\frac{\partial u}{\partial x} = 0, \tag{5}$$

$$\frac{\partial u}{\partial t} + \frac{\partial \eta}{\partial x} + \alpha u = 0. \tag{6}$$

Eliminating $\eta$ from (5) and (6) yields a single-variable wave equation of $u$ as

$$\frac{\partial^2 u}{\partial t^2} + \alpha\frac{\partial u}{\partial t} - 2\frac{\partial u}{\partial x} - x\frac{\partial^2 u}{\partial x^2} = 0. \tag{7}$$

We solve this equation for wave evolution over initially stationary water on a uniform slope given an offshore wave profile of arbitrary form at $x = 1$. For this purpose, we apply Laplace transform to (7). Then, we have

$$(s^2 + \alpha s)\hat{u} - 2\frac{\partial \hat{u}}{\partial x} - x\frac{\partial^2 \hat{u}}{\partial x^2} = 0, \tag{8}$$

where $\hat{u}$ is the Laplace transform of $u$, which is given by

$$\hat{u}(x,s) = \int_0^\infty u(x,t)e^{-st}dt \tag{9}$$

with a complex number frequency parameter $s$. Equation (8) is the modified Bessel's equation and has a bounded solution at the shoreline ($x = 0$) as

$$\hat{u}(x,s) = G(s)x^{-\frac{1}{2}}I_1(2\sqrt{s(s+\alpha)}x^{\frac{1}{2}}), \tag{10}$$

where $I_n$ is the modified Bessel function of the first kind and the $n$th order, and $G(s)$ is an arbitrary function that will be determined by the offshore boundary condition.

Accordingly, the water surface elevation in the Laplace domain, $\hat{\eta}(x,s)$, can be obtained from (5) and (10) such that

$$\hat{\eta}(x,s) = -\frac{1}{s}\left(\hat{u} + x\frac{\partial \hat{u}}{\partial x}\right) = -\frac{\sqrt{s(s+\alpha)}}{s}G(s)I_0(2\sqrt{s(s+\alpha)}x^{\frac{1}{2}}). \tag{11}$$

To determine $G(s)$, we use the boundary condition, $\hat{\eta}(1,s) = \hat{\eta}_0(s)$, where $\hat{\eta}_0(s)$ is the Laplace transform of the observed time series of water surface elevation at $x = 1$. Then, we can express the solutions for water surface elevation and horizontal velocity as

$$\hat{\eta}(x,s) = s\hat{\Psi}_0(x,s)\hat{\eta}_0(s), \tag{12}$$

$$\hat{u}(x,s) = -s\hat{\Psi}_1(x,s)\hat{\eta}_0(s), \tag{13}$$

where $\hat{\Psi}_0(x,s)$ and $\hat{\Psi}_1(x,s)$ are the transfer functions for $\hat{\eta}$ and $\hat{u}$ in the Laplace domain, respectively, which are given as

$$\hat{\Psi}_0(x,s) = \frac{I_0(2\sqrt{s(s+\alpha)}x^{\frac{1}{2}})}{sI_0(2\sqrt{s(s+\alpha)})}, \quad \hat{\Psi}_1(x,s) = \frac{x^{-\frac{1}{2}}}{\sqrt{s(s+\alpha)}}\frac{I_1(2\sqrt{s(s+\alpha)}x^{\frac{1}{2}})}{I_0(2\sqrt{s(s+\alpha)})}. \tag{14}$$

Here, $\hat{\eta}$ and $\hat{u}$ are expressed as the product of each transfer function and $s\hat{\eta}_0$ in the Laplace domain. The factor of $s$ was isolated in (12) and (13) such that the transfer functions are invertible to the time domain. Consequently, $\eta$ and $u$ in the time domain are commonly expressed by the convolution of the rate of water surface displacement at the offshore boundary with the inverse Laplace transforms of the transfer functions as follows.

$$\eta(x,t) = \Psi_0(x,t) * \frac{d\eta_0}{dt}, \tag{15}$$

$$u(x,t) = -\Psi_1(x,t) * \frac{d\eta_0}{dt}, \tag{16}$$

where the asterisk denotes the convolution operation. $\Psi_n(x,t)$ results from the inverse Laplace transform of $\hat{\Psi}_n(x,s)$, which is expressed using the Bromwich integral such that

$$\Psi_n(x,t) = \frac{1}{2\pi i}\int_{\gamma-i\infty}^{\gamma+i\infty}\hat{\Psi}_n(s)e^{st}ds \tag{17}$$

Here, $\gamma$ is a positive real number greater than every singularity of $\hat{\Psi}_n(s)$. The common form of solutions in (15) and (16) suggests that the convolution kernels accommodate the general processes of wave transformation under the observed-wave boundary condition. An exception arises at the shoreline ($x = 0$), where $\hat{\Psi}_1$ is not well defined. The two transfer functions in (14) can be related at the shoreline as $\hat{\Psi}_1(0,s) = s\hat{\Psi}_0(0,s)$, which turns into their relation in time domain as $\Psi_1(0,t) = \partial\Psi_0(0,t)/\partial t$. With this relation, the kinematic condition at the shoreline results from (15) and (16);

$$u_s = -\frac{\partial\xi}{\partial t}, \tag{18}$$

where $u_s(t) \equiv u(0,t)$ represents the shoreline velocity, and $\xi(t) \equiv \eta(0,t)$ expresses the vertical shoreline displacement. This suggests that the choice of the bounded solution at the shoreline in (10) is equivalent to imposing the linearised kinematic condition. It should be emphasised that the present solutions are based on the moving boundary condition at the shoreline. The shoreline velocity can be obtained from (15) and (18).

## 2.2 Kernel derivation

The next step is to derive the kernel functions by evaluating the Bromwich integrals in (17). The residue theorem can be used to determine the inverse Laplace transform of the transfer function. Both $\hat{\Psi}_0(x,s)$ and $\hat{\Psi}_1(x,s)$ have infinite number of poles

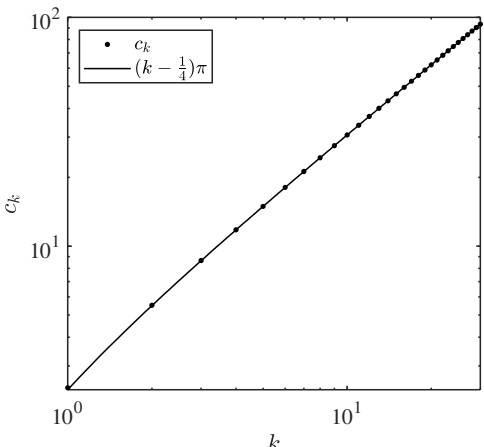

**Figure 2.** Positive zeros of the Bessel function of the first kind and zeroth order; $J_0(c_k) = 0$.

at $s = s_k$ such that $I_0(2\sqrt{s_k(s_k + \alpha)}) = 0$. Additionally, $\hat{\Psi}_0(x, s)$ has a pole at $s = 0$, while the singularity of $\hat{\Psi}_1(x, s)$ at $s = 0$ is removable. The complex zeros of the modified Bessel function are distributed on the imaginary axis; thus, they can be associated with positive real zeros of the Bessel function of the first kind and zeroth order, $c_k$, that satisfies $J_0(c_k) = 0$. All poles of the two functions can be summarised as follows:

$$s_0 = 0, \quad s_k = \frac{-\alpha + i\sqrt{c_k^2 - \alpha^2}}{2} \quad \text{and} \quad \bar{s}_k = \frac{-\alpha - i\sqrt{c_k^2 - \alpha^2}}{2}, \tag{19}$$

where $k$ is a natural number ($k = 1, 2, 3 \cdots$), and $c_k$ is the positive zeros of $J_0$ in ascending order of magnitude, as graphed in Figure 2. Equation (19) suggests that the damping factor has an upper limit such that $\alpha < c_1 \approx 2.405$. When $k$ is large, the positive zeros of the Bessel function can be approximated as

$$c_k \approx (k - \frac{1}{4})\pi. \tag{20}$$

This approximation suggests that $c_k$ linearly increases with $k$, as shown in Figure 2.

Because both functions are holomorphic elsewhere, the Bromwich integrals can be evaluated using the theory of residue such that

$$\Psi_0(x, t) = \text{Res}[\hat{\Psi}_0(s_0)e^{s_0 t}] + \sum_{k=1}^{\infty} \left\{ \text{Res}[\hat{\Psi}_0(s_k)e^{s_k t}] + \text{Res}[\hat{\Psi}_0(\bar{s}_k)e^{\bar{s}_k t}] \right\},$$

$$\Psi_1(x, t) = \sum_{k=1}^{\infty} \left\{ \text{Res}[\hat{\Psi}_1(s_k)e^{s_k t}] + \text{Res}[\hat{\Psi}_1(\bar{s}_k)e^{\bar{s}_k t}] \right\}, \tag{21}$$

where $\text{Res}[\hat{\Psi}(s)e^{st}]$ is the residue of $\hat{\Psi}(s)e^{st}$ at the point $s$. Substituting (14) and (19) into (21) yields

$$\Psi_0(x, t) = 1 + \psi_0(x, t), \quad \Psi_1(x, t) = x^{-\frac{1}{2}}\psi_1(x, t) \tag{22}$$

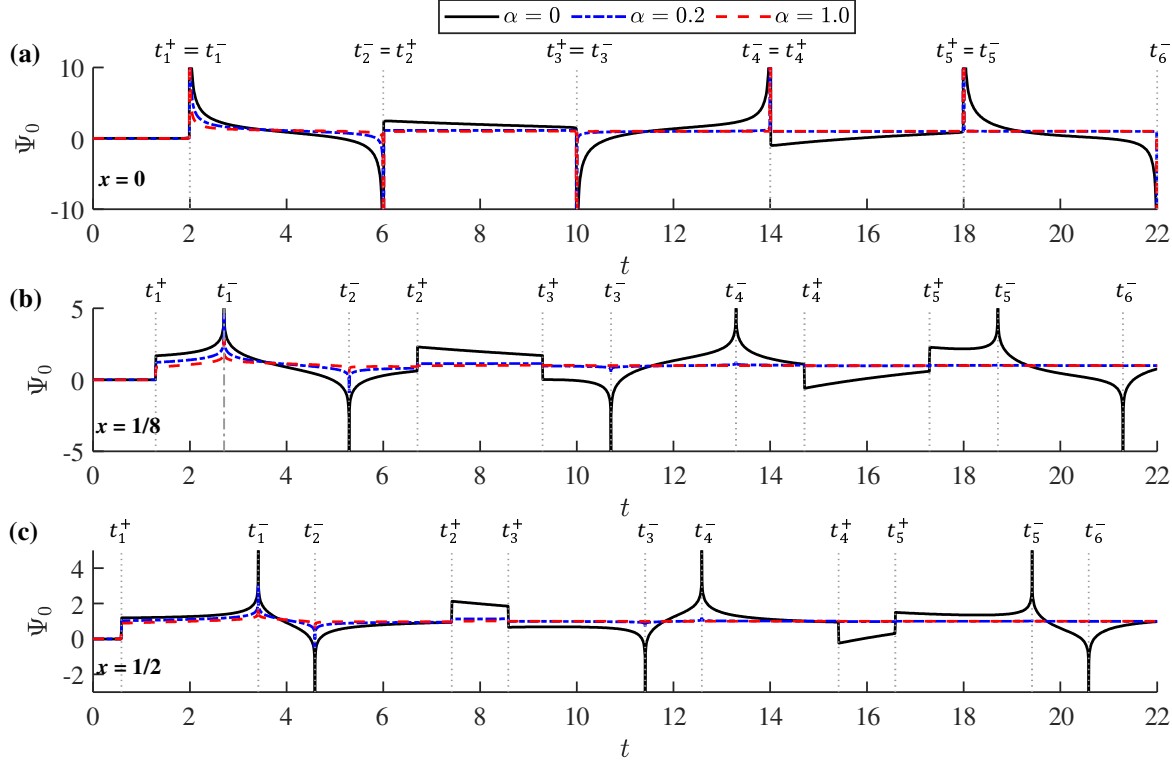

**Figure 3.** Function $\Psi_0(x,t)$ at (a) $x = 0$, (b) $x = 1/8$ and (c) $x = 1/2$. Each panel shows graphs of the function for $\alpha = 0$, 0.2, and 1.0. Vertical dot lines represent the timing of singularities, $t_m^\pm$.

with

$$\psi_n(x,t) = -2e^{-\frac{\alpha t}{2}} \sum_{k=1}^{\infty} \frac{J_n(c_k x^{\frac{1}{2}})}{\lambda_k J_1(c_k)} \sin(\frac{\lambda_k}{2}t + \frac{n\pi}{2} + \theta_k), \quad \lambda_k = \sqrt{c_k^2 - \alpha^2}, \quad \theta_k = \mathrm{atan2}(\lambda_k, \alpha) \tag{23}$$

Both kernels consist of infinite series of the combination of the Bessel function of $x^{\frac{1}{2}}$ and the sinusoidal function of $t$. In addition, the kernels contain the negative exponential function of time and decay with time at a higher rate for larger $\alpha$. For $\alpha = 0$ without the damping effect, the kernels exhibit oscillatory behaviours in the infinite time length; thus, the kernel convolution involves a full wave history from the initial time at the offshore boundary. Introduction of the damping effect confines the causal relation to the near past. Kernel convolution deals with the separation of incident and reflected waves because it is formulated with the observed-wave profile containing two-way components. Separation requires a longer observed-wave history for a smaller damping factor.

Figures 3 and 4 show graphs of $\Psi_0(x,t)$ and $\Psi_1(x,t)$ respectively on the time axis for different values of the damping factor, namely, $\alpha = 0$, 0.2 and 1.0. $\Psi_0(x,t)$ is shown at $x = 0$, 1/8 and 1/2, while $\Psi_1(x,t)$ is graphed at $x = 1/8$, 1/2 and 1. Of note, $\Psi_0(1,t) = 1$ and $\Psi_1(0,t)$ are not well defined. The initial forms of the two kernels agree with those derived under the incident-

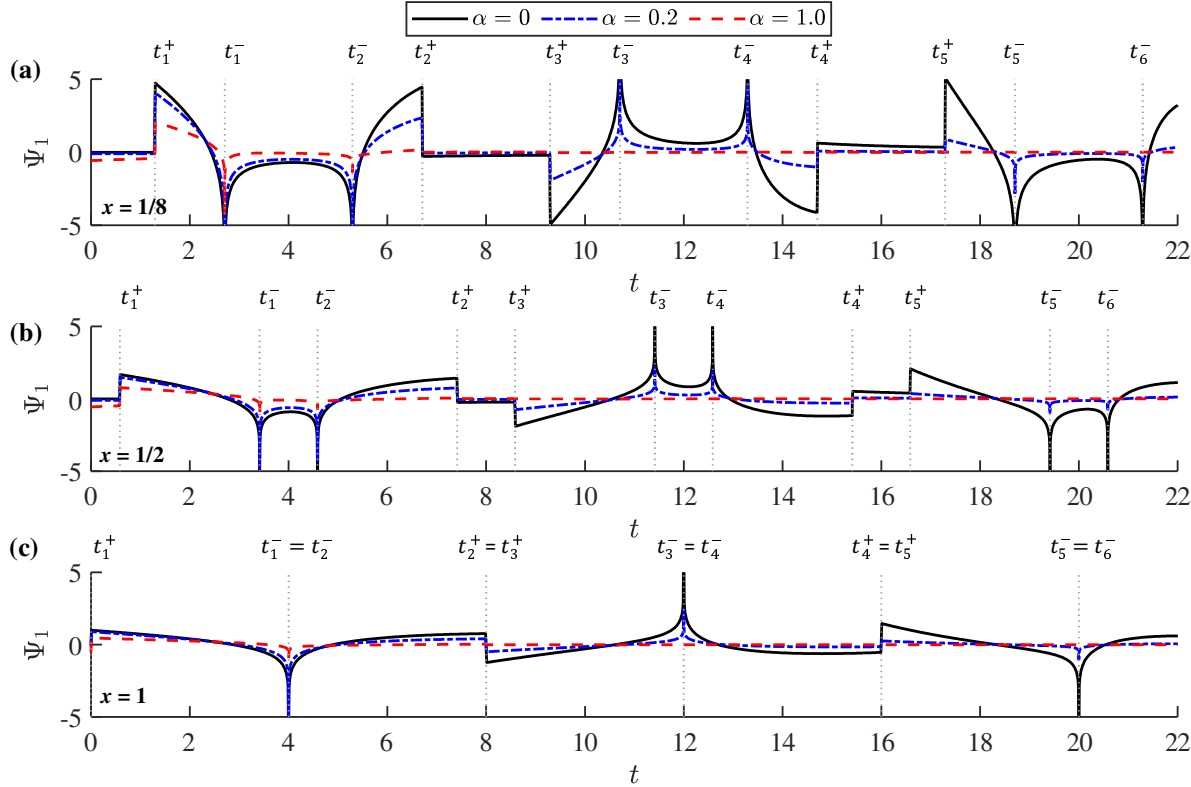

**Figure 4.** Function $\Psi_1(x,t)$ at (a) $x = 1/8$, (b) $x = 1/2$ and (c) $x = 1$. Each panel shows the graphs of the function for $\alpha = 0, 0.2,$ and $1.0$. Vertical dot lines represent the timing of singularities, $t_m^\pm$.

wave boundary condition (Shimozono, 2020) because the initial part of observed wavedata represents a shoreward-propagating wave. They deviate from the incident-wave kernels in the large time domain exhibiting cyclic singularities. Cyclic singularities occur at the following timing:

$$t_m^\pm = 2\left\{1 \pm (-1)^m x^{\frac{1}{2}}\right\} + 4(m-1), \tag{24}$$

where $m$ is the natural number. Both $\Psi_0$ and $\Psi_1$ diverge to positive or negative infinity at $t_m^-$ and form slope discontinuities at $t_m^+$. Of note, $2(1 - x^{\frac{1}{2}})$ corresponds to the propagation time from the offshore boundary to the reference point, whereas $2(1 + x^{\frac{1}{2}})$ represents the propagation time between the two locations via reflection on the shore. Hence, the two kernels have cyclic singularities representing arrivals of past seaward and shoreward-propagating waves with a period of $t = 4$, which equals to the wave round-trip time over the entire slope. Equation (24) can be alternatively described as a series of characteristic curves in the $(x,t)$ plane, $x = \{t - 2 - 4(m-1)\}^2/4$, which represent forth and back propagation of a wave signal on the slope. Kernel convolution generates a coastal waveform by processing the two-way propagating waves at the offshore boundary. At the shoreline ($x = 0$) where $t_m^+ = t_m^-$, the divergence and slope discontinuity merge with each other, and the bimodal kernel

structure disappears from $\Psi_0$. A similar merge of singularities also happens to $\Psi_1$ at the offshore boundary ($x = 1$). The infinite series in (23) is convergent except at $t_m^-$, where it turns into a harmonic series for large $k$; the convergence properties of the infinite series are more rigorously discussed in Appendix A. Therefore, the infinite series can be numerically calculated until the point of convergence except at the singular points.

Substituting (22) into (15) and (16) yields

$$\eta(x,t) = \eta_0(t - t_0) + \int_{t_0}^{t} \psi_0(x,\tau) \left.\frac{d\eta_0}{dt}\right|_{t-\tau} d\tau, \tag{25}$$

$$u(x,t) = -x^{-\frac{1}{2}} \int_{t_0}^{t} \psi_1(x,\tau) \left.\frac{d\eta_0}{dt}\right|_{t-\tau} d\tau, \tag{26}$$

where $t > t_0 \equiv 2(1 - x^{\frac{1}{2}})$ and $\eta = u = 0$ otherwise. These representations suggest that $\eta(x,t) \approx \eta_0(t - t_0)$ and $u(x,t) \approx 0$ hold when the incoming wave is much longer than the slope length. This can be viewed as the seabed slope works like a vertical wall against a relatively long wave. The convolution describes wave deformation over the slope under the observed-wave boundary condition. Despite cyclic singularities, the kernel convolution is well defined, and the convolution can be numerically performed, which will be presented in the next section.

## 3  Kernel convolution

Kernel representations derived in the previous sections allow us to predict coastal waveforms from observed wavedata apart from the shore. The prediction can be made instantly by kernel convolution of the offshore wavedata. This is advantageous over conventional ways of numerically solving the shallow-water equations, which involve both time and spatial integration of the equations over a physical domain. Especially, an accurate simulation of shoreline response requires high-resolution discretisation. Additionally, the numerical model approaches require iterative calculations for the separation of the incident and reflected waves. Despite the efficiency benefits, the numerical convolution requires special treatment for cyclic kernel singularities. Here, an efficient algorithm for the singular kernel convolution is presented and then validated in a simple case of monochromatic-wave propagation.

### 3.1  Numerical method

The kernels have cyclic singularities at $t_m^+$ and $t_m^-$. For the convenience of presenting the convolution method, the singular time points are redefined as follows;

$$t_{jm} = 2\left\{1 - (-1)^j x^{\frac{1}{2}}\right\} + 4(m - 1) \tag{27}$$

where $j$ is 0 or 1 ($t_m^- = t_{0m}$ and $t_m^+ = t_{1m}$). To avoid midpoint singularities, the integration interval is divided into multiple time segments; $[t_{01}, t_{11}], [t_{11}, t_{02}], [t_{02}, t_{12}], \cdots, [t_{0m}, t_{1m}], [t_{1m}, t_{0m+1}], \cdots$. Then, based on the kernel form, different variable

changes are applied in $[t_{0m}, t_{1m}]$ and $[t_{1m}, t_{0m+1}]$ based on the double-exponential integration formula by Takahasi and Mori (1974) via

$$\tau_{0m}(p_0) = 2\left\{2m - 1 + x^{\frac{1}{2}}\tanh\left(\frac{\pi}{2}\sinh p_0\right)\right\} \quad \text{for} \quad [t_{0m}, t_{1m}], \tag{28}$$

$$\tau_{1m}(p_1) = 2\left\{2m + (1 - x^{\frac{1}{2}})\tanh\left(\frac{\pi}{2}\sinh p_1\right)\right\} \quad \text{for} \quad [t_{1m}, t_{0m+1}]. \tag{29}$$

These changes of variables will concentrate integration points of time near the singular points using the double-exponential function. Consequenty, each segment is mapped to an infinite interval of $p_j$ with vanishing endpoint singularities. For shoreline motions ($x = 0$), the segments of $j = 0$ disappear because of $t_{0m} = t_{1m}$. Because the integrand decays with a double-exponential rate, actual numerical integration is performed within a finite interval of $p_j$. Therefore, the convolutions in (25) and (26) are rewritten as

$$245 \quad \eta(x, t) = \eta_0(t - t_0) + \sum_{m=1}^{M}\sum_{j=0}^{1}\int_{-\infty}^{\infty}\psi_0(x, \tau_{jm}(p_j))\left.\frac{d\eta_0}{dt}\right|_{t-\tau_{jm}(p_j)}\frac{d\tau_{jm}}{dp_j}dp_j, \tag{30}$$

$$u(x, t) = -x^{-\frac{1}{2}}\sum_{m=1}^{M}\sum_{j=0}^{1}\int_{-\infty}^{\infty}\psi_1(x, \tau_{jm}(p_j))\left.\frac{d\eta_0}{dt}\right|_{t-\tau_{jm}(p_j)}\frac{d\tau_{jm}}{dp_j}dp_j, \tag{31}$$

where $M$ is an integer ceiling of $(t - t_0)/4$ when $\alpha = 0$ and can be a smaller number when $\alpha > 0$ because the kernels decay on the time axis. The convolution is numerically performed by applying the segment-by-segment integration following the

trapezoidal rule. The convolution is efficiently implemented by the advance tabulation of $\psi_n d\tau_{jm}/dp_j$. We read the kernel table and then perform computation of the infinite integrals in several segments to predict a coastal waveform.

### 3.2    Validation

The convolution method is demonstrated and validated by comparing numerical results with an analytical solution for a simple problem. Here, we consider a simple problem in which a monochromatic wave is observed apart from the shore that contains

both incident and reflected waves. The observed wave is now given as

$$\eta_0(t) = \sin(\lambda t) \tag{32}$$

where $\lambda$ is the dimensionless angular frequency. It is assumed that the coastal wave is in an equilibrium state, and a partial standing wave is formed owing to wave decay over the uniform slope.

    To derive the equilibrium solution, we first take the Laplace transform of (32) and substitute it into (12). Then, we have

$$260 \quad \hat{\eta}(x, s) = \frac{\lambda I_0(2\sqrt{s(s + \alpha)}x^{\frac{1}{2}})}{I_0(2\sqrt{s(s + \alpha)})(s + i\lambda)(s - i\lambda)}. \tag{33}$$

This can be inversely transformed into time domain by the residue theorem. However, the residues associated with the zeros of the modified Bessel function, $s_k$ and $\bar{s}_k$ in (19), yield a transient solution because it accommodates the decaying exponential

function of time with the damping factor $\alpha$. Therefore, we need to consider only the residues of $s = \pm i\lambda$ for the equilibrium solution in case of $\alpha > 0$.

Consequently, the damped equilibrium solution for water surface elevation, $\tilde{\eta}(x,t)$, is given by:

$$\tilde{\eta}(x,t) = a(x)\sin\{\lambda t + \phi(x)\}, \tag{34}$$

where the wave amplitude $a(x)$ and the phase $\phi(x)$ are given by

$$a(x) = \frac{\sqrt{\{AC(x) + BD(x)\}^2 + \{AD(x) - BC(x)\}^2}}{A^2 + B^2}, \quad \phi(x) = \mathrm{atan2}\{AD(x) - BC(x), AC(x) + BD(x)\} \tag{35}$$

with

$$A = \mathrm{Re}\left(I_0(2\sqrt{i\lambda(i\lambda + \alpha)})\right), \quad B = \mathrm{Im}\left(I_0(2\sqrt{i\lambda(i\lambda + \alpha)})\right),$$
$$C(x) = \mathrm{Re}\left(I_0(2\sqrt{i\lambda(i\lambda + \alpha)x})\right), \quad D(x) = \mathrm{Im}\left(I_0(2\sqrt{i\lambda(i\lambda + \alpha)x})\right).$$

The wave amplitude on the slope cannot be represented by simple mathematical functions and exhibits drastic changes with varying $\lambda$ because the observation point moves between the node and anti-node of the partial standing wave.

To validate the numerical convolution method, the wave amplitude is computed for non-zero $\alpha$ values using (30). Because the damping effect confines the causal relation in a finite time, we set the total time length of the convolution such that the kernel value sufficiently decays ($t = 12/\alpha$). The numerical convolution was performed over a sufficiently large interval ($-3 < p_j < 3$) for both $j = 0$ and 1 with a common subinterval $\Delta p_j$. To investigate the sensitivity of the result to $\Delta p_j$, the numerical solutions were obtained with three different subintervals, i.e. $\Delta p_j = 0.05$, 0.1, and 0.25.

Figure 5 shows the comparison of wave amplitude from numerical convolution and the exact solution at three different locations ($x = 0$, 1/2 and 1/8). The two cases with different values of the damping parameter, $\alpha = 0.1$ and 1.0, are shown in the left and right columns, respectively. In each figure, exact and numerical solutions of wave amplitude are compared on the $\lambda$ axis. The dimensionless angular frequency can be expressed as $\lambda = 2\pi l/L_0$ with the representative wavelength $L_0$; thus, $\lambda = 2\pi$ represents the situation in which wavelength is equal to the slope length. In the case of $\alpha = 0.1$, wave amplitude shows large fluctuations on the $\lambda$ axis owing to considerable reflection. The coastal wave amplitude becomes very large when a partial node is formed at $x = 1$ where wave ampliude is fixed to be unity. The fluctuations are smeared out in the case of $\alpha = 1.0$ owing to the stronger damping effect. Numerical results agree well with the exact values in this range of $\lambda$ when $\Delta p_j < 0.1$. Numerical error appears from the high-frequency side when $\Delta p_j = 0.25$. These results confirm that numerical convolution produces an exact solution with a sufficiently small subinterval relative to the tsunami wave period.

In case of $\alpha = 0$, the monocromatic wave forms a complete node at $x = 1$ when $\lambda = c_k/2$. Therefore, we have no wave signal at $x = 1$ in the equilibrium state, and the wave amplitude given by (35) goes to infinity because the boundary value given by (32) is not appropriate. The kernel method works even in such a case as long as the full standing wave is formed from an initially stationary state. To demonstrate this, we consider a transient indicent wave given by

$$\eta_i(t) = \tanh(\frac{\lambda t}{10})\sin(\lambda t), \tag{36}$$

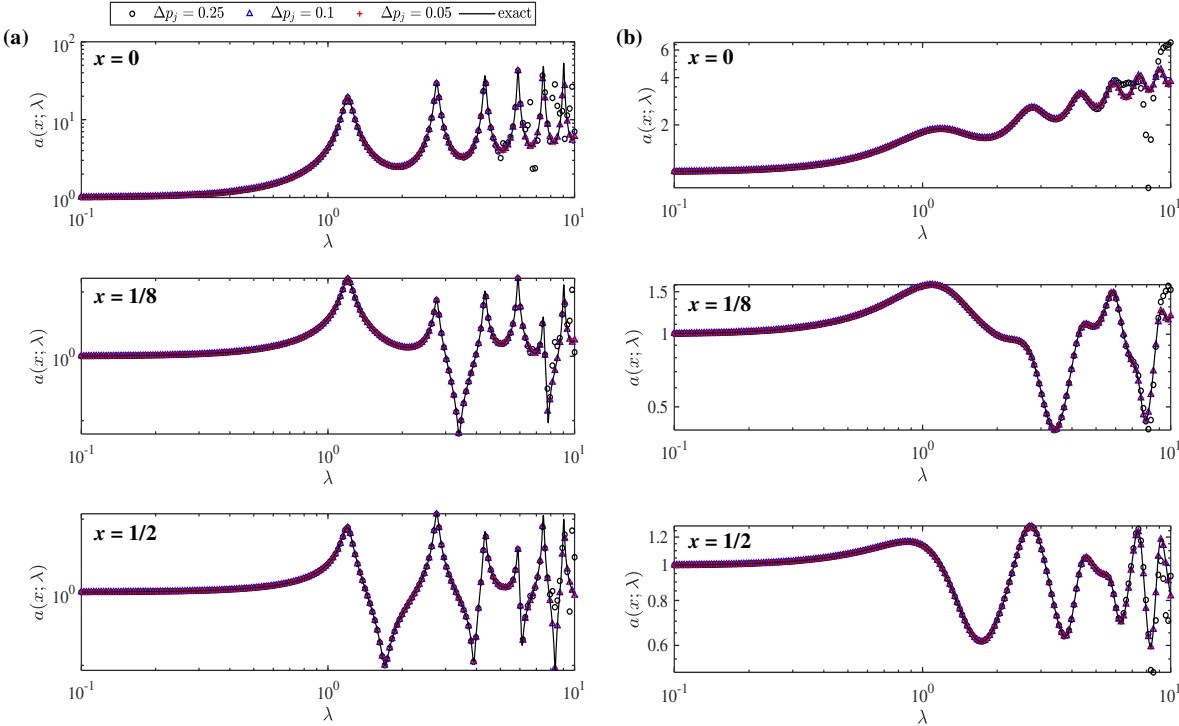

**Figure 5.** Wave amplitude $a$ as a function of dimensionless angular frequency $\lambda$ at different locations ($x =0$, 1/8 and 1/2) for damping coefficients (a) $\alpha = 0.1$ and (b) $\alpha = 1.0$. Each panel shows exact and numerical solutions with different integration subintervals $\Delta p_j = 0.05$, 0.25 and 0.1.

where the tanh function was introduced for a smooth transition from the stationary water surface to a monochromatic wave. As illustrated in Figure 6a, the incident wave is growing to a monochromatic wave over several wave periods. When we set $\lambda = c_1/2 \approx 1.2024$, a node will be formed at $x = 1$ under the incident wave. Using the incident-wave kernel previously proposed by the author (Shimozono, 2020), we can obtain the water surface elevation at $x = 1$, $\eta_0$, as shown in Figure 6b. The null elevation datum occurs after the initial transient phase due to the formation of a full standing wave. Figure 6c compares shoreline elevation profiles, $\xi$, computed from $\eta_i$ using the incident-wave kernel and that computed from $\eta_0$ using the present kernel. The two results show a perfect agreement, and the wave amplitude coverges to the analytical solution of the monochromatic wave runup height by Keller and Keller (1964), $2/\sqrt{J_0(c_1)^2 + J_1(c_1)^2} \approx 3.85$. Even if the null datum occurs at $x = 1$ as a result of superposition of incident and reflected waves, the kernel convolution can predict a waveform at any location on the slope from the initial transient part of the wave history at $x = 1$. It is worth emphasising that the present kernel works for such a problem because it is constructed for the intial-boundary value problem. Without the initial condition, we could not derive an incident wave signal from offshore wave data when a full node is formed at the boundary.

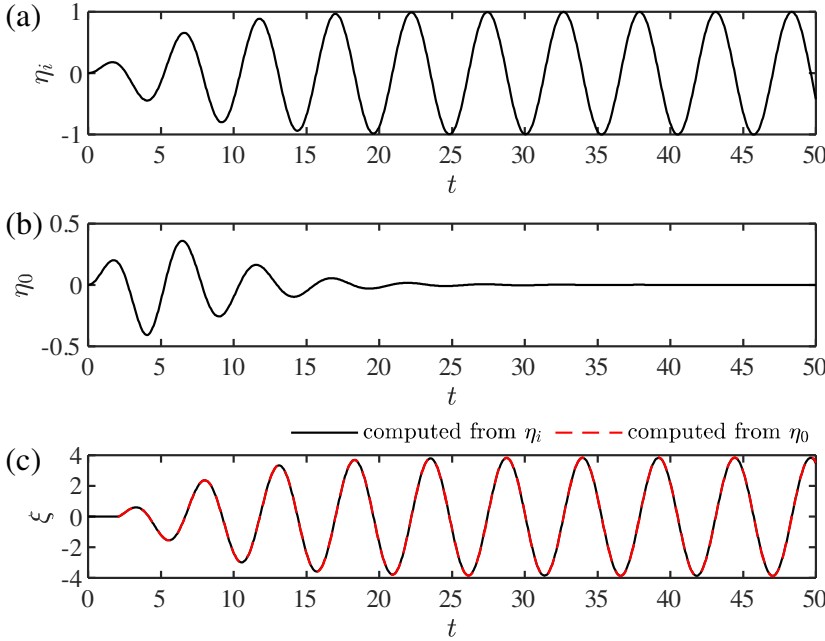

**Figure 6.** Kernel validation for the formation of a full standing wave ($\lambda = c_k/2$): (a) incident wave, $\eta_i(t)$; (b) observed wave at $x = 1$, $\eta_0(t)$, and (c) shoreline elevation data computed from $\eta_i(t)$ and $\eta_0(t)$.

## 4 Kernel applications

We can instantly predict a coastal waveform over the slope from the offshore wave profile via the kernel convolution. To demonstrate its capability in real-world tsunami problems, the kernel method is applied to cases of the 2011 Tohoku tsunami, for which instrumentally recorded wavedata are available at different locations along the affected coastline (Kawai et al., 2013). Based on the availability of multiple wavedata in the same area, two cases were selected on the northern side of the epicentre as shown in Figure 7a. Case 1 is located far north of the tsunami source, but tsunami run-up on the coast still reached several metres above the mean sea level. The offshore wavedata is available relatively close to the coast, and the coastal tide station also recorded the water surface fluctuation due to the tsunami. In contrast, Case 2 is located in one of the most devastated parts of the coastline, where three offshore stations were aligned perpendicularly to the coast. Located close to the tsunami source, the initial wave was dominant over successive waves, and the tsunami run-up reached up to 20 m in this area (Shimozono et al., 2012). Despite the lack of a coastal wave record, this case is employed owing to the availability of multiple observations in the cross-shore direction. The leading part of the tsunami at each observed location is assumed to have been refracted in the deep water and approached the coastline nearly perpendicularly. This assumption may not be valid for the most offshore point of Case 2 (TM1), where water depth exceeds 1,500 m and the depth contour is significantly curved. To apply the kernel method, the water depth is assumed to linearly decrease over the distance between the observed location and the shore in each case,

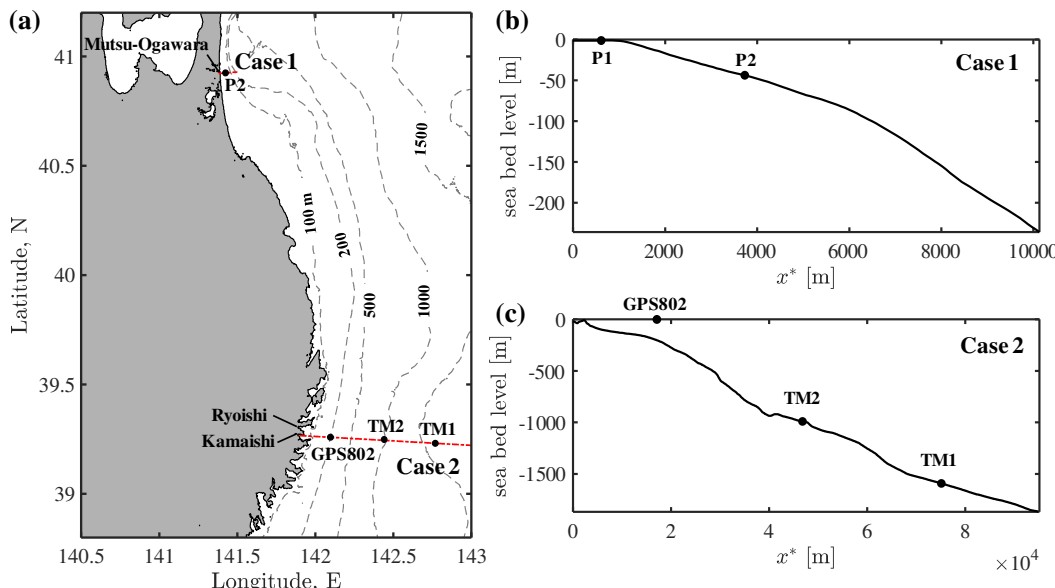

**Figure 7.** Map of the case study sites and wave observation stations for the 2011 Tohoku tsunami: (a) overview of the two case study sites at the northeastern Pacific coast of Japan; (b) and (c) seabed profiles along the crosshore transects for Case 1 and 2, respectively. The markers indicate the locations of wave observation points.

even though the actual seabed slope is not strictly uniform. The cross-shore bathymetry and locations of wave observation in each case are shown in Figures 7b and 7c.

### 4.1 Case 1:Mutsu-Ogawara

Case 1 is located at the coast of Mutsu-Ogawara, where relatively small waves of similar amplitude repeatedly attack the coast at the period of approximately 30 min. The offshore wavedata were recorded by an ultrasonic wave gauge installed on the seabed at a depth of 43.8 m, approximately 3.4 km off the coast (P1 in Figure 7b). In addition, an onshore wave record is available from a tide station inside the Mutsu-Ogawara Port (P2 in Figure 7b). Because the offshore wavedata contained high-frequency fluctuations that may not be real waves propagating shoreward, the following analyses were performed with the low-pass filtered time series ($<$0.003 Hz) as shown in Figure 8a. The numerical convolution was performed with the filtered wavedata with three different values of the damping parameter, $\alpha = 0.1$, 0.5, and 1.0, on the assumption of a uniform slope ($\beta = 43.8/3400 \approx 0.013$). The numerical integration in each segment was implemented for $-3 < p_j < 3$ with $\Delta p_j = 0.05$. The numerical error was confirmed to be negligibly small.

Figures 8b–c compare the computed waveforms at the shoreline, $\xi^*$, and the observed wavedata at P2 with the different values of damping parameters, respectively. The first term in (25), $\eta_0^*(t - t_0)$, is also plotted to highlight the slope effect described by kernel convolution. The comparison of $\xi^*$ and $\eta_0^*$ shows that relatively short wave components are significantly amplified

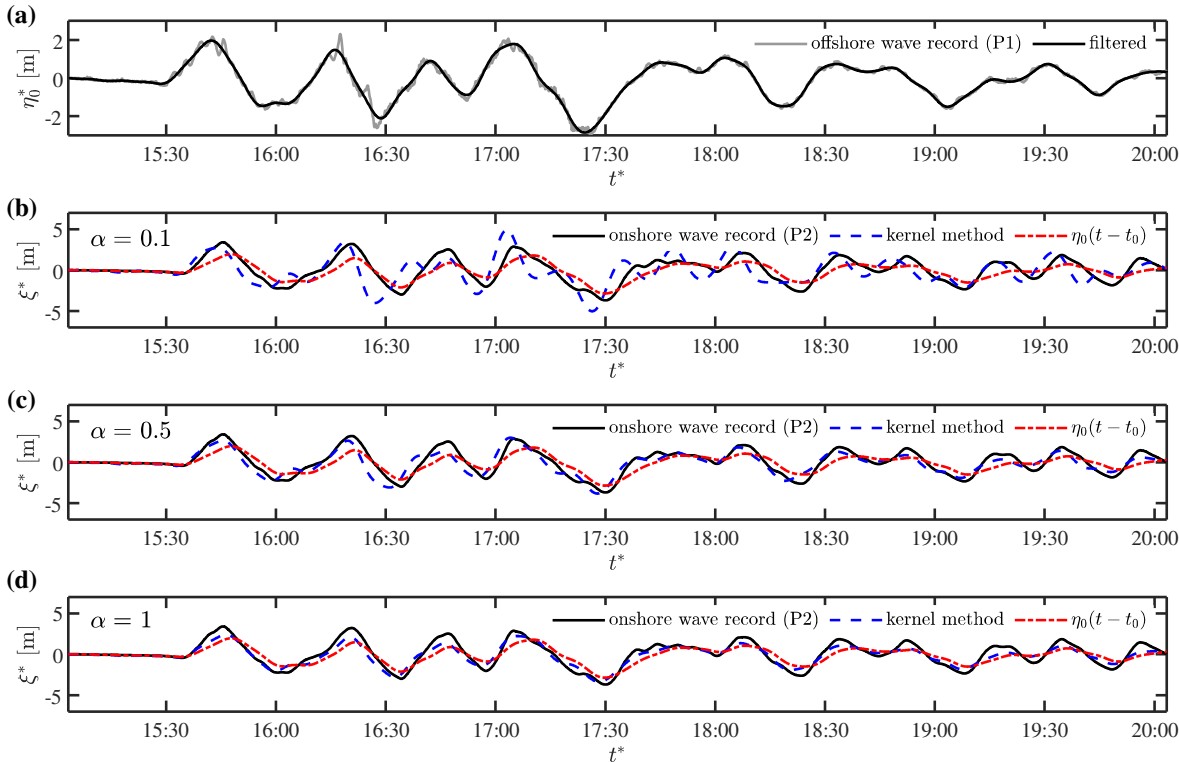

**Figure 8.** Prediction of the shoreline motions in Case 1: (a) observed and low-pass filtered wavedata at P1; (b), (c), and (d) waveforms at the shoreline computed with $\alpha$ = 0.1, 0.5, and 1.0, respectively, in comparison to the coastal wave record at P2.

over the slope. The computed results with $\alpha = 0.1$ overestimate the short waves on the shore. The observed wavedata at P2 are in good agreement with the computed result of $\alpha = 0.5$, but the initial waveforms are reproduced slightly better with $\alpha = 1.0$. This result suggests that the damping factor varies with wave amplitude, and the non-linear formulation of the damping term may be required to achieve higher accuracy. Nonetheless, despite the crude assumptions, the kernel method works well to predict the coastal waveform from the observed data a few kilometres off the coast when the $\alpha$ value is appropriately chosen.

### 4.2 Case 2: Kamaishi and Ryoishi

Case 2 is located on the coast ranging from Kamaishi to Ryoishi, which are highly devastated areas during the 2011 tsunami event. Two bottom pressure sensors (TM1 and TM2) captured the initial waveform of the deep-water tsunami propagating towards the coast (Maeda et al., 2011; ERI, 2011). As shown in Figure 7, TM1 was located 76 km off the coast at a depth of approximately 1600 m, while TM2 was 47 km off the coast at a depth of approximately 1000 m. In addition, a GPS-buoy (GPS802) deployed 15 km off the coast recorded a full profile of the tsunami propagating at a depth of approximately 200 m (Kawai et al., 2013). Because three observation points are aligned on a cross-shore line, the dataset can be used to validate

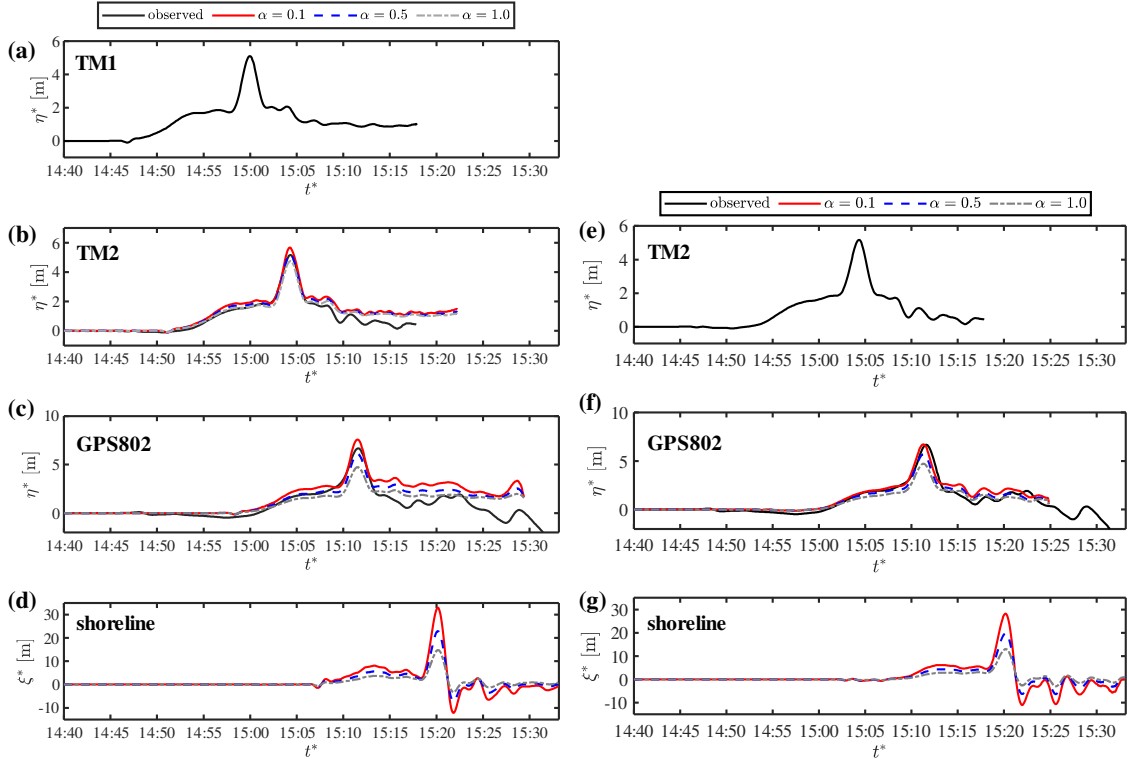

**Figure 9.** Prediction of coastal waveforms in Case 2: (a) observed wavedata at TM1; (b) predicted waveforms based on TM1 and observed wave at TM2; (c) predicted waveforms based on TM1 and observed wave at GPS802; (d) predicted shoreline motion based on TM1; (e) observed wavedata at TM2; (f) predicted wave based on TM2 and observed wave at GPS802; (g) predicted shoreline motion based on TM2.

the kernel method for the long-distance propagation of the real-world tsunami with a large amplitude. Coastal wave gauges in shallow water were destroyed by the tsunami; thus, there is no data for onshore waveforms. Nevertheless, intensive post-event surveys provide the range of coastal run-up heights in this area (Shimozono et al., 2012). The ria coast exhibits an intricate coastline, but the dimension of coastline variations is smaller than the tsunami wavelength. Therefore, tsunami propagation is expected to be described with the one-dimensional kernel approach to a large extent.

Figure 9a shows the observed waveform at TM1. The leading part of the tsunami in this region was characterised by a short, impulsive wave of large amplitude riding on a relatively long wave. We predict the waveforms at the locations of TM2 and GPS803 via the kernel convolution of the wavedata at TM1, assuming the water depth to be linearly decreasing from TM1 to the shore ($\beta = 0.02$). Figure 9b and 9c compare the computed waveform for $\alpha = 0.1$, 0.5, and 1.0 with the observed data at TM2 and GPS802, respectively. The computed profile agrees with the observed one at both TM1 and GPS802 when $\alpha = 0.5$. However, the water surface elevation is predicted to be higher after the peak at both locations, and this discrepancy occurs probably because the observed wave at TM1 is located far from the coast and contains wave components that do not propagate shoreward. The resulting run-up height on the shore is sensitive to the choice of the damping parameter, as shown in Figure

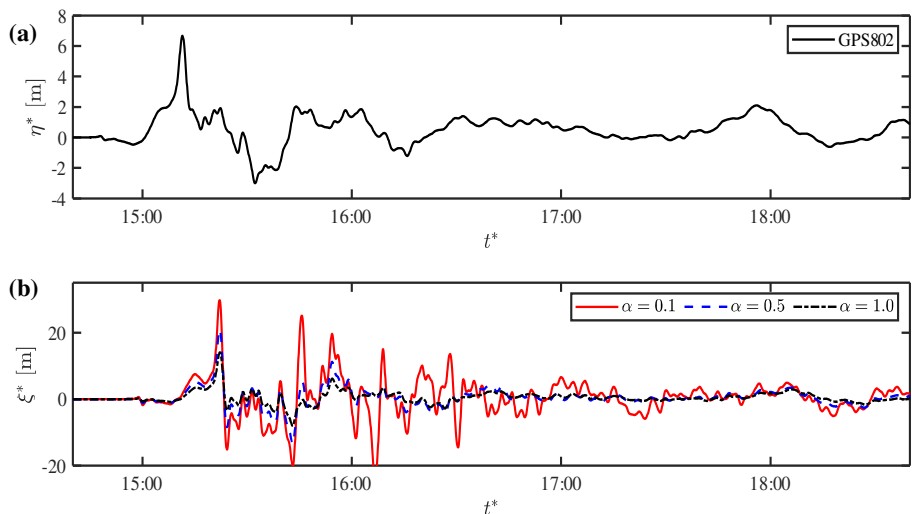

**Figure 10.** Prediction of shoreline motions in Case 2: (a) observed wavedata at GP802; (b) predicted shoreline motion based on GPS802.

9d. The measured run-up heights after the event significantly varied owing to the intricate coastline; however, the maximum run-up height was up to 20 m in the coastal areas. Therefore, the $\alpha$ value of 0.5-1.0 should be employed for the purpose of predicting the run-up of the tsunami in this area.

Next, we predict the waveform at the location of GPS803 using the wavedata at TM2 graphed in Figure 9e. The computed results at GPS802 show better agreement with the observation than those predicted from TM1, as shown in Figure 9f. The results in 9c and 9f confirm that the observed wave at TM1 did not fully propagate shoreward and suggest that TM1 is too far from the shore for the kernel application. The lower value of $\alpha$ realises better agreement at GPS802, but the maximum run-up height is overestimated with the same value (Figure 9c). This poses the limitation of the linear formulation of the damping effect that cannot account for relatively high tsunami attenuation in shallow water owing to the quadratic dependence on flow velocity. In particular, the local topographic elements along the intricate coastline caused additional damping effects in the coastal area. Therefore, the empirical damping parameter should be optimised for the target location of the prediction. Figure 10 shows another prediction of wave profile at the shoreline from the wavedata at GPS802 that was available for a much longer period than TM2. The shoreline displacement shows a similar form to the previous predictions, and the low damping parameter of $\alpha = 0.1$ produces successive high waves that are not observed in this area. The onshore waveform of the leading wave in the case of $\alpha$ = 0.5–1.0 agrees well with the previous model result based on the two-dimensional non-linear shallow water equations (Shimozono et al., 2012).

The kernel representation of coastal tsunami evolution provides an efficient method for predicting onshore tsunami waveforms based on a single wave observation apart from the shore. Coastal waveforms are almost simultaneously obtained with the observed profile because the computation time for the numerical integration is negligibly small compared to the tsunami time scale. In this case, the lead times for the real-time prediction of shoreline motions are 16 min and 8 min when predicted

from TM2 and GPS802, respectively. The prediction accuracy is limited by the crude assumptions underlying kernel representation that makes the fast prediction possible: one-dimensional propagation over a uniform slope with the linear damping effect. Nonetheless, the damping parameter can be optimised for the prediction of tsunamis at a specific coastal location from an offshore observation point; this can be done with the help of a numerical model. The two cases suggest that coastal run-up is well-predicted when the damping parameter is set in the range of 0.5–1.0.

## 5 Conclusions

This study presents the tsunami propagation kernel that compactly accommodates damped propagation processes of tsunamis over the coastal slope. The kernel representation was derived under the observed-wave boundary condition unlike the previous kernels based on an incident-wave boundary condition. Therefore, it can be directly applied to predict a coastal waveform from observed wavedata that contain both shoreward and seaward-propagating components. Furthermore, the damping factor was incorporated into the kernel to collectively represent tsunami attenuation that occurs in various ways. Both the water surface elevation and horizontal flow velocity over the slope can be represented as the kernel convolution of the rate of water surface displacement apart from the shore. Kernel functions have cyclic singularities to separate the two-way propagating wave in the offshore wave history. The introduction of the damping factor confines this causal relation in the short past. The kernel convolution can be efficiently performed via the change of variables using the double-exponential function, and the numerical convolution method worked satisfactorily if the integration subinterval was chosen to be sufficiently small relative to the tsunami wave period.

Kernel representation has potential applicability to real-world problems because it can instantly predict the response of coastal water to incoming wave trains in deep water. It can be applied for the real-time prediction of coastal waveforms from a single offshore observation by bottom pressure-type or buoy-type wavemeters that are widely deployed off the coasts under an imminent threat of tsunamis. In addition, it can be incorporated into deep-water tsunami simulations to efficiently predict onshore waveforms without resolving coastal bathymetry. Prediction accuracy depends on the choice of the damping parameter because the linear damping term cannot fully represent actual tsunami decay that occurs in diverse ways. Therefore, the damping parameter should be treated as an empirical parameter and can be optimised at each target site through pre-calibration based on numerical simulations. Even though the number of cases is limited, the damping parameter of 0.5–1.0 works for different cases to reasonably reproduce onshore waveforms. Additional studies will be needed to confirm the applicability of semi-empirical kernel and to improve kernel formulation for better representation of tsunami decay in shallow water.

*Data availability.* The observation data of the 2011 Tohoku tsunami are available from the website of the Nationwide Ocean Wave Information Network for Ports and Harbour (https://nowphas.mlit.go.jp/pastdatars/data/NOWPHAS_Tsunami_data.zip).

## 410  Appendix A: Kernel convergence properties

This appendix presents the convergence properties of the infinite series in (23). Firstly, we define the sequence of the series as

$$f_{nk}(x,t) = \frac{J_n(c_k x^{\frac{1}{2}})}{\lambda_k J_1(c_k)} \sin(\frac{\lambda_k}{2} t + \frac{n\pi}{2} + \theta_k). \tag{A1}$$

Then, the infinite series can be rewritten as

$$\psi_n(x,t) = -2e^{-\frac{\alpha t}{2}} \sum_{k=0}^{\infty} f_{nk}(x,t) \tag{A2}$$

We look into the behavior of the sequence for large $k$. When $k$ is large, we can use the approximation of $c_k$ in (20) and the following approximations as well;

$$\lambda_k \approx c_k \approx (k - \frac{1}{4})\pi \quad \theta_k \approx \frac{\pi}{2} \tag{A3}$$

Furthermore, the Bessel functions can be approximated base on their asymptopic expansion for a large augument

$$J_n(z) \approx \sqrt{\frac{2}{\pi z}} \cos(z - \frac{n\pi}{2} - \frac{\pi}{4}), \tag{A4}$$

which is valid only for large $z$. Since the augment of the Bessel function in the numerator of (A1) contains $x^{\frac{1}{2}}$, we discuss the property of the sequence separately in two cases; (a) $x = 0$ and (b) $x > 0$.

**(a)** $x = 0$

Here we discuss the behavior of only $f_{0k}(0,t)$ because $f_{1k}(0,t)$ is not well defined. Since $J_n(0) = 1$, we can approximate 425  (A1) for large $k$ as

$$f_{0k}(0,t) \approx \frac{(-1)^{k-1}}{\sqrt{2k - \frac{1}{2}}} \cos\left\{\frac{1}{2}\left(k - \frac{1}{4}\right)\pi t\right\} \tag{A5}$$

The sequence decays with increasing $k$ while oscillating due to the combination of the alternating sign and cosine function of $k$. Therefore, the infinite series of the sequence is generally convergent. However, the oscillations disappear when $t = t_m^{\pm} = 4m - 2$ such that

$$f_{0k}(0,t_m^{\pm}) \approx \frac{-1}{\sqrt{2k - \frac{1}{2}}} \cos\left\{\frac{\pi}{4} - \frac{m\pi}{2}\right\} \tag{A6}$$

This sequence forms a general harmonic series, and thus, the kernel slowly diverges to positive or negative infinity depending on the $m$ value at $t = t_m^{\pm}$ as shown in Figure 3a.

**(b)** $x > 0$

When $x > 0$, the sequence can be approximated for large $k$ as

$$f_{nk}(x,t) \approx \frac{(-1)^k}{\pi x^{\frac{1}{4}}\left(k - \frac{1}{4}\right)} \cos\left\{\pi\left(k - \frac{1}{4}\right)x^{\frac{1}{2}} - \frac{n\pi}{2} - \frac{\pi}{4}\right\} \cos\left\{\frac{\pi}{2}\left(k - \frac{1}{4}\right)t + \frac{n\pi}{2}\right\} \tag{A7}$$

This sequence also exhibits both oscillation and decay with increasing $k$, and thus, the infinite series is convergent except at singular points where the oscillations cease. Substituing $t = t_m^{\pm}$ into (A7) and separting it into two parts, we have

$$
\begin{aligned}
f_{nk}(x, t_m^{\pm}) \approx & \frac{1}{2\pi x^{\frac{1}{4}}\left(k - \frac{1}{4}\right)} \cos\left[\pi\left(k - \frac{1}{4}\right)x^{\frac{1}{2}}\left\{1 \pm (-1)^m\right\} - \frac{m\pi}{2}\right] \\
& + \frac{(-1)^n}{2\pi x^{\frac{1}{4}}\left(k - \frac{1}{4}\right)} \sin\left[\pi\left(k - \frac{1}{4}\right)x^{\frac{1}{2}}\left\{1 \mp (-1)^m\right\} + \frac{m\pi}{2}\right].
\end{aligned}
\tag{A8}
$$

The orginal sequence is now expressed as sum of the two sequences. When $t = t_m^+$, either sequence vanishes and the other one forms a convergent series. Therefore, the infinite series is convergent at $t = t_m^+$. On the other hand, either sequence forms a convergent series and the other one generates a harmonic series when $t = t_m^-$. This confirms that the infinite series in (A2) slowly diverges to positive and negative infinity alternatively at $t_m^-$ as shown in Figure 3b-c and 4.

*Author contributions.* The author confirms sole responsibility for study conception and design, analysis and manuscript preparation.

*Competing interests.* I declare I have no competing interests.

*Acknowledgements.* This study was supported by Grant-in-Aid for Scientific Research No. 18H01541 from Japan Society for the Promotion of Science.

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
