# Peer review of "Tsunami propagation kernel and its applications"

_Natural Hazards and Earth System Sciences, 2021_

## Referee Comment (RC2)

Report on the manuscript

**Tsunami propagation kernel and its applications**

by

Takenori Shimozono

The proposed manuscript deals with the prediction of the coastline inundation through the use of proper analytical kernels for the Linear Shallow Water Equations. The basic idea has been already developed in a previous paper of the author (i.e. Shimozono 2020) and is further inspected here in order to define a more straightforward procedure for the data assignment. In particular, the formulation proposed allows one to assign a boundary datum that includes both incident- and reflected-wave components. In addition, the author proposes the use of a damping factor which accounts for the energy dissipation encountered by the wave during its path toward the shoreline.

My overall opinion is that the manuscript is well written and that the mathematical approach is sound. Further, the proposed analytical solution is compared with measurements from real tsunami and this makes the present contributions interesting for the readers of NHESS. There are, in any case, some aspects that deserve an accurate inspection and a deeper insight. Below, I list the points that have to be addressed before the paper may be accepted.

**Major Points**

- The author states that the proposed solution applies to a generic datum (that is, a signal that includes both reflected and incident components). In any case, differently from his previous work (Shimozono 2020), he does not provide any evidence of this.

  My personal opinion is that this statement by the author is not correct and this is confirmed by the occurrence of a singular kernel in the equation (14) as a consequence of the assignment in (11). Indeed, in absence of dissipation (namely, $\alpha = 0$), the complex values of $p$ such that $I_0(2s) = 0$

correspond to a null wave elevation datum at the seaward boundary (while in general $G(s) \not\equiv 0$). This occurs when the reflected and incident components of the wave elevation are in opposite phase at $x = 1$. In the same case, the velocity at $x = 1$ is generally different from zero [see the equation (10)].

This points has to be clarified with care. I think that the cause of the presence of singularities in the kernel is simple due to the assignment on variables which do not exclusively represent the whole incident signal.

- The presence of the dissipation (i.e. $\alpha \neq 0$) seems to mitigate somehow the singularity of the kernel, since the poles are not aligned along the imaginary axis [see the equation (19)]. I think some comments about the differences between the inviscid and viscous solutions should be added.

- Section 3.2. As shown in the previous sections, the singularities in the kernel are "accounted for" through the integration along the Bromwich path. This means that the singularities are handled through principal-value integrals. This is a well established procedure. What I do not understand is why the solution for $\eta$ in the equation (33) is obtained by substituting the equation (32) inside the equation (12) straightforwardly. The solution that is obtained is obviously ill-posed and cannot be accepted in this form.

  This is clear if we observe the Figure 5. For example the panel *(a)* shows an amplitude which is $\mathcal{O}(10^2)$ while the amplitude at the seaward limit is $\mathcal{O}(1)$! More in general, the overall behaviour of the amplitude described in this figure is quite odd and seems not physical.

  Once again, I stress that the occurrence of singularities in the solution for $\eta$ is here mitigated by the presence of the damping factor [see the expressions for $A$ and $B$ just after the equation (35)]. In fact, if $\alpha = 0$, we would obtain a singular solution of $\eta$ for those values of $\lambda$ such that $I_0(2i\lambda) = 0$.

  In conclusion, I think that all this section has to be rewritten by deriving a solution for $\eta$ obtained through the use of principal-value integrals (as done in the previous sections).

**Minor Points**

- Page 6, line 144. Maybe $\hat{\Psi}_1$ instead of $\Psi_1$.

- Page 6, line 154. The author states "*Both $\Psi_0(x, s)$ and $\Psi_1(x, s)$ have a pole at $s = 0 \ldots$*". Actually, it seems to me that $\Psi_1(x, s)$ has a removable singularity at $s = 0$ [ I refer to the second formula in the equation (14)]. Please check again.

- I think that the expressions for $t^{\pm}$ in the equation (24) are simply the two branches of some specific characteristic curves in the $(x, t)$-plane. Indeed they may be cast in the following compact form:

$$x = [\, t - 2 - T\,(m-1)\,]^2 \,, \tag{1}$$

  where $T = 4$ is the time "period" that takes a signal to travel back and forth in the fluid region.

- Equation (19). Here the author should point out that the damping factor has an upper-limit. Specifically, it should be $\alpha \leq c_1$ where $c_1 \simeq 2.405$ is the first real zero of the Bessel function $J_0$.

**References**

T. Shimozono, *Kernel representation of long-wave dynamics on a uniform slope*, Proc. R. Soc. A 476: 20200333.

---

## Author Comment (AC1)

**Responses to comments from Prof. Efim Pelinovsky**

I appreciate the reviewer for his comments to reinforce the manuscript. I will revise the manuscript with valuable inputs.

**General comment**

The paper under review is devoted to an interesting problem of the connection between the moving shoreline fluctuations and the recording of sea level fluctuations at a fixed point (tide-gauge). Usually such a connection with the incident wave characteristics is considered, but in this case, the tide-gauge record is the superposition of the incident and reflected waves. The obtained solution is important for recalculating the available tide gauge records into moving shoreline fluctuations but with strict constraints on the coastal zone geometry (the rectangular channel, the linearly inclined beach). The solution is obtained strictly within the linear theory framework taking the linear friction into account. Then it is applied to the analysis of the 2011 Tohoku tsunami. I have no objections to the reviewed paper, only a few minor comments.

**Response: I appreciate the reviewer for the positive evaluation of the manuscript.**

**Individual comments**

1. The authors correctly note that "it is well known that the wave amplitude is not significantly affected by non-linearity unless the non-linear wave distortion leads to wave breaking". I would like to add that in the linearly inclined bottom case, it is easy to recalculate the results within the framework of the linear theory for nonlinear moving shoreline oscillations (if there is no wave breaking). I would also like to note that the maximum runup characteristics important for practice turn out to be identical in the linear and nonlinear theory. This fact is noted in several works cited by the author, in particular in the paper (Pelinovsky & Mazova, 1992). I think, it should be mentioned in the reviewed paper as it will reinforce the importance of the linear results.

**Response: I agree with the reviewer that the importance of linear solutions should be more stressed by adding the suggested points. I will revise the manuscript accordingly.**

2. The author justifies the linear damping introduction only by the need for an analytical solution. Meanwhile, in tsunami practice, this term is relatively widely used, see, for example, the latest work (Davies G, Romano F and Lorito S Global Dissipation Models for Simulating Tsunamis at Far-Field Coasts up to 60 hours Post-Earthquake: Multi-Site Tests in Australia. Front. Earth Sci. 2020, vol. 8: 598235. Doi: 10.3389 / feart.2020.598235) and references therein. The references to such works are sure to improve the transition from the theoretical work and tsunami practice.

**Response: I appreciate the valuable information. In the revised manuscript, I will describe the previous linear damping applications in tsunami modeling, which would support the current approach.**

3. A long time ago the paper by Mazova, R.Kh., Osipenko, NN, and Pelinovskiy, Ye.N. "A dissipative model of the runup of long waves on shore" (Oceanology, 1990, vol. 30, N. 1, 29 – 30) was published. In the above-mentioned work, the same linear shallow water equations were solved, only a monochromatic incident wave was considered as an input. It is worth referring to in reviewed manuscript.

**Response: I was not aware of the work, and I will consider referring to it in the revised manuscript.**

4. I would like to note the confusion in the list of references. No pages are indicated in the papers of Chan & Liu and Didenkulova et al. The paper by Choi et al seems to be mixed with some other paper (therefore, the authors and pages should be checked).

**Response: I am sorry for the confusion. I will correct the reference list.**

---

## Author Comment (AC2)

**Responses to comments from Reviewer #2**

I appreciate the reviewer for the in-depth review. The thoughtful comments and feedback would help improve the manuscript. Each comment has been carefully considered point by point. I hope that I did not mistake the reviewer's points.

**General comment**

The proposed manuscript deals with the prediction of the coastline inundation through the use of proper analytical kernels for the Linear Shallow Water Equations. The basic idea has been already developed in a previous paper of the author (i.e. Shimozono 2020) and is further inspected here in order to define a more straightforward procedure for the data assignment. In particular, the formulation proposed allows one to assign a boundary datum that includes both incident- and reflected-wave components. In addition, the author proposes the use of a damping factor which accounts for the energy dissipation encountered by the wave during its path toward the shoreline.

My overall opinion is that the manuscript is well written and that the mathematical approach is sound. Further, the proposed analytical solution is compared with measurements from real tsunami and this makes the present contributions interesting for the readers of NHESS. There are, in any case, some aspects that deserve an accurate inspection and a deeper insight. Below, I list the points that have to be addressed before the paper may be accepted.

**Response: I appreciate the reviewer for the positive evaluation and for raising critical points to improve the manuscript.**

**Major comments**
1. The author states that the proposed solution applies to a generic datum (that is, a signal that includes both reflected and incident components). In any case, differently from his previous work (Shimozono 2020), he does not provide any evidence of this.

   My personal opinion is that this statement by the author is not correct and this is confirmed by the occurrence of a singular kernel in the equation (14) as a consequence of the assignment in (11) Indeed, in absence of dissipation (namely, $\alpha=0$), the complex values of p such that $I0(2s) = 0$ correspond to a null wave elevation datum at the seaward boundary (while in general

G(s) ≠ 0). This occurs when the reflected and incident components of the wave elevation are in opposite phase at x = 1. In the same case, the velocity at x=1 is generally different from zero see the equation (10).

This points has to be clarified with care. I think that the cause of the presence of singularities in the kernel is simple due to the assignment on variables which do not exclusively represent the whole incident signal.

**Response: I had the same concern as the reviewer during this work, but the statement in the manuscript is correct. The kernel was formulated with a generic datum, $\eta_0(t)$, that contains both incident and reflected components. In the case of $\alpha = 0$, the complex values of $s$ such that $I_0(2s) = 0$ corresponds to complex frequencies at which a node is formed at x=1, i.e. a null wave elevation as stated by the reviewer. I interpreted the reviewer's concern as the incident wave signal may not be derived from the generic datum at x=1 for such frequency components. Here I would like to stress that the kernel was formulated with an initial condition (stationary water) using the Laplace transform. Therefore, in principle, we could get the incident wave signal from $\eta_0(t)$ by following back its history to the initial state (This is what the kernel convolution does).**

**The present kernel was derived through mathematically rigorous procedures without any approximation. The resulting formulation suggests that it is possible to get $\eta(x,t)$ from $\eta_0(t)$ through the principal-value integral despite the singularities. Please see my reply to Comment 3, in which I newly demonstrate a critical case supporting that the kernel perfectly works even when a node is formed at x=1.**

2. The presence of the dissipation (i.e. $\alpha \neq 0$) seems to mitigate somehow the singularity of the kernel, since the poles are not aligned along the imaginary axis [see the equation (19)]. I think some comments about the differences between the inviscid and viscous solutions should be added.

**Response: The presence of the dissipation does not essentially change the singular kernel structure. Indeed, the singular structure remains in the kernel of $\alpha \neq 0$. The shift of poles by $\alpha$ from the imaginary axis introduces a decaying exponential of time into the kernel. This**

**means that we only need a shorter history of $\eta_0$ to isolate the incident wave and then obtain wave solutions over the slope. The dissipation changes the causal relation and makes the kernel more compact in time (we do not need to look back to the initial state). It was explained in the original manuscript as the differences between the inviscid and viscous cases. This difference can be interpreted as "mitigation" in a practical sense, but I believe it is not what the reviewer means.**

3. Section 3.2. As shown in the previous sections, the singularities in the kernel are "accounted for" through the integration along the Bromwich path. This means that the singularities are handled through principal-value integrals. This is a well established procedure. What I do not understand is why the solution for $\eta$ in the equation (33) is obtained by substituting the equation (32) inside the equation (12) straightforwardly. The solution that is obtained is obviously ill-posed and cannot be accepted in this form.

   This is clear if we observe the Figure 5. For example the panel (a) shows an amplitude which is $O(10^2)$ while the amplitude at the seaward limit is $O(1)$! More in general, the overall behaviour of the amplitude described in this figure is quite odd and seems not physical.

   Once again, I stress that the occurrence of singularities in the solution for $\eta$ is here mitigated by the presence of the damping factor [see the expressions for A and B just after the equation (35)]. In fact, if $\alpha = 0$, we would obtain a singular solution of $\eta$ for those values of $\lambda$ such that $I0(2i\lambda) = 0$.

   In conclusion, I think that all this section has to be rewritten by deriving a solution for $\eta$ obtained through the use of principal-value integrals (as done in the previous sections).

**Response: The case of monochromatic waves in 3.2 might be a bit confusing, but the results are physically correct. In this case, a monochromatic wave of unit amplitude is observed at x=1 as a result of a superposition of incident and reflected waves. Therein, we look at waves in an equilibrium (steady) state neglecting the initial transient phase, namely, partial standing waves over the slope. The solution can be obtained by inverting Eq. (33) into the time domain using the residue theorem. For the equilibrium solution under the viscous condition, we do not need to consider the residues associated with zeros of the modified Bessel function which form a transient part of the solution vanishing with time.**

The odd behaviors of the wave amplitude in the figure are because a node is formed at x=1 due to the superposition at some frequencies. In the case of small dissipation, the ratio of wave amplitudes at x=0 to x=1 becomes very large because the node of the partial standing wave is formed at x=1, as shown in the top panel of Figure 5a. Also, the node could be formed on the slope where the ratio becomes very small, as shown in the lower panels of Figure 5a. Furthermore, if $\alpha = 0$, the ratio goes to infinity at $\lambda = c_k/2$ when a complete node is formed due to a full standing wave. This is what the reviewer pointed out, and this occurs because the amplitude is fixed to be unity at x=1 (the amplitude is not known a priori, especially for viscous cases).

In response to the comments by the reviewer, including previous ones, I show a critical case of $\alpha = 0$ as evidence for the kernel works even when a node is formed at x=1. Now we consider a transient monochromatic incident wave given by

$$\eta_i(t) = \tanh(\frac{\lambda t}{10}) \sin(\lambda t) \tag{2}$$

As illustrated in Figure 1a, the incident wave is growing to a monochromatic wave from an initially stationary state. The tanh function was introduced for a smooth transition from the stationary state to a monochromatic wave during several wave periods. When we set $\lambda = c_1/2 = 1.2024$, a node will be formed at x=1. Figure 1b shows $\eta_0 = \eta(1,t)$ computed from $\eta_i$ using the incident-wave kernel proposed in my previous paper (Shimozono, 2020). The null elevation datum occurs after the initial transient phase due to the formation of a full standing wave. Figure 1c compares the shoreline elevation, $\xi$, computed from $\eta_i$ using the incident-wave kernel and that computed from $\eta_0$ using the present kernel. The two results show a perfect agreement, and the wave amplitude approaches the analytical solution of the monochromatic wave runup, $2/\sqrt{J_0(c_1)^2 + J_1(c_1)^2} \approx 3.85$. Even if the null datum occurs at $\lambda = c_k/2$, the kernel convolution can predict a waveform at any location over the slope from the initial transient part of the wave history at x=1. Therefore, this case supports my previous statement above that the formulation is NOT mitigated by the presence of dissipation. (The reviewer's concern is true if we formulate the kernel for waves in the infinite time domain using Fourier transform.)

In the revised manuscript, I will revise Section 3.2 for clarity and explain the raised point more clearly with the new case shown here.

[Figure]

Figure 1 Kernel validation for a monochromatic incident wave starting from the stationary water: (a) incident wave, $\eta_i(t)$, (b) observed wave at x=1, $\eta_0(t)$, and (c) shoreline elevation data computed from $\eta_i(t)$ and $\eta_0(t)$

**Minor comments**

4.  Page 6, line 144. Maybe $\widehat{\Psi}_1$ instead of $\Psi_1$.

**Response: I will correct it.**

5.  Page 6, line 154. The author states "Both $\Psi_0(x, s)$ and $\Psi_1(x, s)$ have a pole at s = 0…". Actually, it seems to me that $\Psi_1(x, s)$ has a removable singularity at $s = 0$ [ I refer to the second formula in the equation (14)]. Please check again.

**Response: The reviewer is right. I will correct it.**

**Reference:**

**Shimozono T. 2020: Kernel representation of long-wave dynamics on a uniform slope, Proc. R. Soc. A.476, 20200333**

---

## Referee Report (RR1)

Report on the manuscript

**Tsunami propagation kernel and its applications**

by

Takenori Shimozono

I agree with the reply made by the author in the revised version. Below, I just list some minor points that have to be addressed.

**Minor Points**

- Following the suggestion by the Referee # 1, the author added the following statements (lines 77-80): "*However, it is well known that the wave amplitude is not significantly affected by non-linearity unless the non-linear wave distortion leads to wave breaking (Carrier and Greenspan, 1958; Tuck and Hwang, 1972; Synolakis, 1991). The nonlinear shoreline motion can be readily derived from the linear solution via the hodograph transform, and the run-up height is unchanged from the linear case (e.g. Pelinovsky and Mazova, 1992).*

  The last sentence is inexact and has to be modified. The linear and non-linear theory give the same extrema at he shoreline when the boundary assignment in the hodograph space is linearized. In this case the hodograph transformation essentially reduces to a map that deforms the linear solution (without affecting the extrema). These are, in fact, the hypotheses under which the largest number of analytical works available in the literature are obtained.

  On the contrary, if we consider the whole boundary assignment (that is, we include the nonlinear contributions), the wave height predicted by the nonlinear theory is larger than the linear theory. An evidence of this is given in Antuono and Brocchini (2007) where the nonlinear contributions are accounted for (at least at the first order of a perturbation approach).

  Please add some comments about this point.

- Again about the difference between linear and nonlinear solutions, it is worth noting that the inclusion of nonlinear contributions substantially modifies the conditions for wave breaking (see, for example, Antuono & Brocchini 2008, Antuono & Brocchini 2010).

  Since the analytical solution proposed in the paper holds true for non-breaking waves, I think that some comments about the range of validity (namely, the range for the occurrence of non-breaking waves) should be added in the revised manuscript.

- Section 3.2. I appreciated the reply by the author. Specifically, he pointed out how the mixed data assignment (initial/boundary data) is substantially different from a boundary data assignment: *"Therefore, this case supports my previous statement above that the formulation is not mitigated by the presence of dissipation. (The reviewer's concern is true if we formulate the kernel for waves in the infinite time domain using Fourier transform.)"*

  I think that a brief comment about this point should be added in the revised manuscript.

**References**

Antuono M. & Brocchini M. *The Boundary Value Problem for the Nonlinear Shallow Water Equations*, Studies in Applied Mathematics, 119: 73-93, (2007)

Antuono M. & Brocchini M. *Maximum run-up, breaking conditions and dynamical forces in the swash zone: a boundary value approach*, Coastal Engineering 55 (2008) 732-740

Antuono M. & Brocchini M. *Solving the nonlinear shallow-water equations in physical space*, J. Fluid Mech. (2010), vol. 643, pp. 207-232.

---

## Author Response (AR2)

**Responses to comments from Reviewer #2**

I appreciate the editor and reviewers for the re-assessment of the manuscript. I revised the manuscript according to the minor comments from Reviewer #2.

**Minor Comments**

1. Following the suggestion by the Referee # 1, the author added the following statements (lines 77-80): "*However, it is well known that the wave amplitude is not significantly a ected by non-linearity unless the non-linear wave distortion leads to wave breaking (Carrier and Greenspan, 1958; Tuck and Hwang, 1972; Synolakis, 1991). The nonlinear shoreline motion can be readily derived from the linear solution via the hodograph transform, and the run-up height is unchanged from the linear case (e.g. Pelinovsky and Mazova, 1992).*

   The last sentence is inexact and has to be modified. The linear and nonlinear theory give the same extrema at he shoreline when the boundary assignment in the hodograph space is linearized. In this case the hodograph transformation essentially reduces to a map that deforms the linear solution (without affecting the extrema). These are, in fact, the hypotheses under which the largest number of analytical works available in the literature are obtained.

   On the contrary, if we consider the whole boundary assignment (that is, we include the nonlinear contributions), the wave height predicted by the nonlinear theory is larger than the linear theory. An evidence of this is given in Antuono and Brocchini (2007) where the nonlinear contributions are accounted for (at least at the first order of a perturbation approach). Please add some comments about this point.

2. Again about the difference between linear and nonlinear solutions, it is worth noting that the inclusion of nonlinear contributions substantially modifies the conditions for wave breaking (see, for example, Antuono & Brocchini 2008, Antuono & Brocchini 2010). Since the analytical solution proposed in the paper holds true for nonbreaking waves, I think that some comments about the range of validity (namely, the range for the occurrence of non-breaking waves) should be added in the revised manuscript.

**Response to Comments 1 and 2 : I agree with the reviewer. The newly added sentence in response to Reviewer #1 was not exact. In order to correct the sentence while keeping the input from Reviewer #1, I modified the paragraph as follows. I clarified that the statement is valid under the linearlized boundary value assignment, but stated that the nonlinear modification is small when the boundary is placed in deep water. I also added some sentences about wave breaking in response to Comment 2 citing the suggested references. However, it is not possible to provide the range of the occurrence of non-breaking waves in general, without specifying a wave type. Therefore, I just stated that the wave breaking criterion can be given as a breakdown point of the hodograph transform given the specific wave condition.**

*However, it is well known that the wave amplitude is not significantly affected by non-linearity unless the non-linear wave distortion leads to wave breaking (Carrier and Greenspan, 1958; Tuck and Hwang, 1972; Synolakis, 1991). The nonlinear shoreline motion can be readily derived from the linear solution via the hodograph transform, and the run-up height is unchanged from the linear case, if the boundary-value assignment is linearized (e.g. Pelinovsky and Mazova, 1992). Furthermore, the ocurrence of wave breaking, which limits the applicable range of the present approach, can be predicted as a breakdown point of the hodograph transform under the same condition. While the linearized boundary-value assignment potentially affects the run-up height and the wave breaking condition, nonlinear modifications are minor as long as the ratio of wave amplitude to water depth is small at the boundary (Antuono and Brocchini, 2007, 2008). Therefore, the main process of practical interest can be described by the linear equations when we place the boundary in deep water.*

3. Section 3.2. I appreciated the reply by the author. Specifically, he pointed out how the mixed data assignment(initial/boundarydata) is substantially different from a boundary data assignment: ”*Therefore, this case supports my previous statement above that the formulation is not mitigated by the presence of dissipation. (The reviewer's concern is true if we formulate the kernel for waves in the infinite time domain using Fourier transform.)*” I think that a brief comment about this point should be added in the revised manuscript.

**Response to Comments 3: I added a brief comment on this point in the end of 3.2 as follows.**

*It is worth emphasising that the present kernel works for such a problem because it is constructed for the intial-boundary value problem. Without the initial condition, we could not derive an incident wave signal from offshore wave data when a full node is formed at the boundary.*